# Mitogen and Stress-Activated Kinases 1 and 2 Mediate Endothelial Dysfunction

**DOI:** 10.3390/ijms22168655

**Published:** 2021-08-11

**Authors:** Naveed Akbar, Calum Forteath, Muhammad S. Hussain, Kathleen Reyskens, Jill J. F. Belch, Chim C. Lang, Ify R. Mordi, U Bhalraam, J. Simon C. Arthur, Faisel Khan

**Affiliations:** 1The Institute of Cardiovascular Research, Ninewells Hospital and Medical School, University of Dundee, Dundee DD1 9SY, UK; c.forteath@dundee.ac.uk (C.F.); m.s.z.hussain@dundee.ac.uk (M.S.H.); j.j.f.belch@dundee.ac.uk (J.J.F.B.); u.bhalraam@dundee.ac.uk (U.B.); 2Division of Cell Signalling and Immunology, School of Life Sciences, University of Dundee, Dundee DD1 5EH, UK; katreyskens@hotmail.co.uk (K.R.); j.s.c.arthur@dundee.ac.uk (J.S.C.A.); 3Division of Molecular and Clinical Medicine, Ninewells Hospital and Medical School, University of Dundee, Dundee DD1 9SY, UK; c.c.lang@dundee.ac.uk (C.C.L.); i.mordi@dundee.ac.uk (I.R.M.)

**Keywords:** endothelium, vascular biology, vascular disease, MAPK, cytokine

## Abstract

Inflammation promotes endothelial dysfunction, but the underlying mechanisms remain poorly defined in vivo. Using translational vascular function testing in myocardial infarction patients, a situation where inflammation is prevalent, and knock-out (KO) mouse models we demonstrate a role for mitogen-activated-protein-kinases (MAPKs) in endothelial dysfunction. Myocardial infarction significantly lowers mitogen and stress kinase 1/2 (MSK1/2) expression in peripheral blood mononuclear cells and diminished endothelial function. To further understand the role of MSK1/2 in vascular function we developed in vivo animal models to assess vascular responses to vasoactive drugs using laser Doppler imaging. Genetic deficiency of MSK1/2 in mice increased plasma levels of pro-inflammatory cytokines and promoted endothelial dysfunction, through attenuated production of nitric oxide (NO), which were further exacerbated by cholesterol feeding. MSK1/2 are activated by toll-like receptors through MyD88. MyD88 KO mice showed preserved endothelial function and reduced plasma cytokine expression, despite significant hypercholesterolemia. MSK1/2 kinases interact with MAPK-activated proteins 2/3 (MAPKAP2/3), which limit cytokine synthesis. Cholesterol-fed MAPKAP2/3 KO mice showed reduced plasma cytokine expression and preservation of endothelial function. MSK1/2 plays a significant role in the development of endothelial dysfunction and may provide a novel target for intervention to reduce vascular inflammation. Activation of MSK1/2 could reduce pro-inflammatory responses and preserve endothelial vasodilator function before development of significant vascular disease.

## 1. Introduction

Inflammation is a major driver in the development of atherosclerosis [1,2,3]. It promotes early endothelial dysfunction [4] as well as playing an important role in ischemia reperfusion following acute myocardial infarction (AMI). Blood cytokines are predictive of future adverse cardiovascular events [5,6,7] and local inflammation in tissues is associated with tissue injury and immune cell recruitment.

Cytokine release can be induced through numerous pathways. Importantly, toll-like receptors (TLRs) induce the expression of pro-inflammatory genes that are associated with atherosclerosis [8]. TLRs are also upregulated on peripheral blood monocytes following AMI [9] and on activated endothelial cells [10,11]. The regulation of cytokine expression downstream of TLRs is complex and involves recruitment of myeloid differentiation primary response-88 (MyD88), initiating downstream signals through multiple signalling pathways, including p38α mitogen-activated protein kinase (MAPK). Inhibition of p38 in a model of atherosclerosis reduces plaque formation [12] and preserves cardiac function in a model of heart failure [13]. However, clinical trials of p38 inhibition have shown no therapeutic effect on cardiovascular disease (CVD) outcome [14].

While p38α has inflammatory roles, recent work shows it can also have anti-inflammatory functions [15]. For example, mice with a myeloid specific deletion in p38α show increased pathology in an ultraviolet-B induced skin inflammation model [16]. The effects of p38 in inflammation are in part mediated by two groups of down-stream kinases, mitogen and stress kinase (MSK) 1 and 2 and MAPK-activated protein kinases (MAPKAPK) 2 and 3. MSK1 and 2 have been found to be important in mediating the anti-inflammatory functions of p38α [16,17]. 

MSK1 and 2 are nuclear kinases that are activated downstream of p38α MAPK and serve to help regulate the induction of immediate early genes [18]. TLR signalling in macrophages has been shown to activate MSKs and this promotes transcription of several genes with anti-inflammatory functions including interleukin-10 (IL-10) and IL-1 receptor antagonist (IL-1ra) [16,17,19]. MSKs also regulate the induction of a dual specificity phosphatase (DUSP1) [16], which is involved in deactivation of p38α and c-Jun N-Terminal kinases (JNK) downstream of TLR signalling [20]. Consequently, MSK1/2 double knockout (KO) mice show increased sensitivity to liposaccharide (LPS) induced endotoxic shock, which is accompanied by increased induction of tumour necrosis factor (TNF), interleukin-6 (IL-6) and IL-12 relative to wild-type (WT) animals [16]. In contrast to MSKs, MAPKAPK2/3 are important in mediating the pro-inflammatory effects of p38α. MAPKAPK2 deficient animals display significant attenuation of TNF-α production in response to LPS [21], resistance to the development of rheumatoid arthritis (RA) [22], and a reduction in atherosclerotic lesion formation [23]. However, the role of these kinases in the development of early vascular inflammation and endothelial dysfunction remains unknown.

The aim of the present study was to examine the role of MSKs in CVD. AMI is associated with substantial impairment of endothelial function and inflammation in both the infarcted myocardium and myocardium perfused by normal vessels [24,25]. The determinants of endothelial dysfunction that contribute to the extent of ischemia and necrosis after AMI are not fully elucidated. It is not known if MSK1/2 are involved in endothelial dysfunction following AMI. In this study, we examined patients presenting with AMI, and obtained peripheral blood mononuclear cells (PBMCs) for evaluation of MSK1/2 expression and to determine the association of this expression with endothelial function in vivo. We hypothesized that there is a link between expression of MSK1/2 and endothelial dysfunction. To examine this relationship in more detail, we used an in vivo MSK1/2 KO mouse model to longitudinally assess the functional role of MSK1/2 in development of endothelial dysfunction, a known precursor for CVD development. Additionally, we used MyD88 and MAPKAPK2/3 deficient mouse models to examine proteins upstream and downstream of p38α to better understand the role of innate immune signalling in the development of endothelial dysfunction in vivo.

## 2. Materials and Methods

### 2.1. Human Cohort of ST-Elevation Myocardial Infarction (STEMI) Patients

We recruited 35 patients who were admitted to hospital with STEMI and under-went percutaneous coronary intervention (PCI) at Ninewells Hospital Coronary Care Unit. Patients provided written, informed consent for the collection of bloods for the isolation of PBMCs and for endothelial function testing within 12–100 h post-PCI investigation conformed to the principles outlined in the Declaration of Helsinki and local ethical approval was granted by review boards at the University of Dundee. A further 28 age-matched healthy volunteers were recruited as controls (Appendix A). Exclusion criteria were <18 years of age, unable to give written consent and undergoing current cancer treatments. Additional exclusion criteria for healthy volunteers, included medical history of vascular diseases such as peripheral arterial disease, stroke, hypertension, and haematological conditions such as hypercoagulability and deep venous thrombosis. A sample size of 40 subjects in each group was projected to provide >90% power at a 5% level of significance to detect a 15% difference in gene expression between groups post-PCI and at least 90% power and 5% significance to detect a 2% difference in flow-mediated dilatation (FMD). 

### 2.2. Assessment of Endothelial Function in Humans—Flow Mediated Dilatation

Endothelial function was assessed by measuring FMD of the brachial artery using an 8–15 MHz linear array ultrasound probe (Sequoia 512, Siemens, Camberley, UK) as described previously [26,27,28]. We assessed the endothelium-independent response by following administration of 0.4 mg sublingual glyceryl trinitrate (GTN). Additionally, for assessing microvascular function, we obtained blood flow velocity measurements using Doppler ultrasound at baseline and after 5 min of upper arm arterial occlusion as described previously [29].

### 2.3. Isolation of Peripheral Blood Mononuclear Cells (PBMCs) RNA

PBMCs were isolated from whole blood using density gradient centrifugation over Ficoll-Paque (GE-Health Care Life Sciences, Illinois, United States). PBMCs were stored in fetal bovine serum (FBS) and dimethyl sulphide (DMSO) (10%) at −80 °C. 

### 2.4. RT-qPCR

#### 2.4.1. RNA Isolation and cDNA Synthesis

Isolation of RNA from PBMCs was performed using the guanidium-thiocyanate phenol-chloroform TRIzol (ThermoFisher, Waltham, United States) method. Briefly, frozen samples of PBMCs were thawed before centrifugation at 1000 rpm for 5 min at 4 °C. Freezing medium was carefully removed from the pelleted PBMCs and the cells were resuspended in cold PBS and centrifuged at 1000 rpm for 5 min at 4 °C. This step was repeated once more before removing the final PBS wash and adding 1 mL of TRIzol solution. Cells were lysed by pipetting and left at room temperature for 5 min. Then, 200 µL of chloroform was added to the TriZol-cell mixture and vigorously mixed. Samples were then rested at room temperature for 4 min before a centrifugation at 13,000 rpm for 15 min at 4 °C. Centrifugation of this mix produces phase separation of the sample, from which the upper aqueous phase containing the RNA was removed and transferred to fresh tube. To the collected aqueous phase, 500 µL isopropanol was added before a brief mix using a vortexer. The mixture was then left to stand at room temperature for 10 min to allow the RNA to precipitate. The RNA was then pelleted by centrifugation at 13,000 rpm for 10 min at 4 °C. The isopropanol supernatant was then removed and replaced by 1 mL of a 75% ethanol wash and vortexed. A subsequent centrifugation at 10,600 rpm for 5 min at 4 °C permitted the collection of an RNA pellet. The ethanol wash was removed and the samples were briefly dried by evaporation in a fume hood before resuspension in 20 µL DEPC-treated deionised water. RNA concentrations were then determined using a Nanodrop-8000 Spectrophotometer (ThermoFisher, Waltham, MA, USA) according to manufacturer guidance. RNA was deemed acceptance for cDNA synthesis when the ratio of absorbance at 260:280nm was 2.0 or greater.

#### 2.4.2. cDNA Synthesis 

cDNA was reversed transcribed using the Superscript II (First strand) cDNA synthesis kit (ThermoFischer) according to the supplier’s instructions. Briefly, RNA samples were diluted using DEPC-deionised water to 100 ng per sample before the addition of a master mix containing 1 μL random primers and 1 μL deoxynucleoside triphosphate (dNTPs) per sample. Samples were then heated to 65 °C for 5 min before being transferred to wet ice. A second mastermix containing 4 μL 5× First-Strand buffer (FSB) (ThermoFischer) and 2 μL Dithiothreitol (DTT) (ThermoFisher, Waltham, United States) was added per sample before a 2 min incubation at room temperature. Superscript II Reverse Transcriptase. (ThermoFischer) (1 µL) was then added to the mixture before incubation at 42 °C for 50 min. A final incubation at 72 °C for 15 min terminated the reaction. 

#### 2.4.3. TaqMan RT-PCR

Real-Time PCR was performed using TaqMan gene expression assay probes from ThermoFisher (Waltham, United States) (Hs01046591_m1 (MSK1), Hs01071879_m1 (MSK2) and Hs01060665_g1 (ACTB) and TaqMan 2× Universal PCR Mastermix (Applied Biosystems) on a QuantStudio™ 7 Flex qPCR machine. (ThermoFischer Scientific Inc., Waltham, United States). cDNA samples were diluted with DEPC-treated deionised water to a concentration of 12.5 ng. PCR conditions were as follows; 10 min hold at 95 °C for DNA denaturation followed by 40 cycles, where samples are reheated to 95 °C for 15 s and cooled to 60 °C for 1 min. Cycle thresholds (CT) was obtained from the qRT-PCR machine for each gene target. The average CT values were obtained from the ΔΔCT method, using ACTB as a ‘housekeeping’ gene. Fold change was determined using the ^2-^(ΔΔCt) method. 

### 2.5. Mouse Models

All experiments were performed under UK Home Office Licences and conducted according to the Animals Scientific Procedures Act 1986 (UK) and directive 2010/63/EU of the European Parliament guidelines on the protection of animals used for scientific purposes. All experiments were approved by the Institutional Animal Welfare and Ethical Review Body. All animals were euthanized in accordance with local, national UK, and EU guidelines.

All mice were male and aged-matched to the WT controls. WT and MSK1/2 double KO [30], MyD88 KO [31] and MAPKAP 2/3 double KO mice [21] were obtained from breeding facilities at the University of Dundee. All mice were backcrossed onto C57Bl6/J (Charles River Laboratories, Bristol, UK) mice for a minimum of 12 generations.

### 2.6. Animal Husbandry and Group Allocation

Adult (10–16 weeks of age) mice were housed in groups of up to six. Diets consisted of standard rodent chow (Special Diets Services (SDS), Essex, UK): Mice were fed diet numbers 1 or 3 and sterilized distilled water fed ad libitum, unless otherwise stated. Animals were transferred from a barrier breeding facility to the experimental facility (in which the necessary equipment was installed) at least one week before vascular function testing, to allow acclimation and to avoid stress. Animals were randomly allocated into four groups: WT control mice were fed a normal rodent chow, WT mice on a cholesterol diet (TD.01383 diet, Harlan-Teklad; 18% protein rodent chow with added cholesterol 2% by weight), and KO mice on normal rodent chow and a cholesterol diet as detailed above. Researchers were blinded to genotypes in the group allocations during vascular assessments for the study duration.

### 2.7. Assessment of Endothelium-Dependent Responses

Skin microvascular responses were measured longitudinally in the same animals at 4-week intervals following baseline measurements using laser Doppler imaging (LDI) as described previously [32,33]. Animals were anaesthetized using 5% Isoflurane (Abbot Laboratories, Chicago, United States) in oxygen (2 L/min), which was delivered using a standard Boyle’s Apparatus and maintained by delivering 1.5–2% Isoflurane via a nose cone for the duration of the procedure. Iontophoresis chambers were initially filled with ~2 mL of 1% solution of the α1-adrenergic receptor agonist phenylephrine (PE) (Sigma-Aldrich, St. Louis, United States). Following constriction, PE was washed out of the chamber with deionized water and replaced with a 2% solution of the endothelium-dependent vasodilator acetylcholine (ACh) (Sigma-Aldrich, St. Louis, United States). ACh was iontophoresed for 10 min using an anodal current of 100 µA and the maximum vasodilator response measured by LDI. 

### 2.8. Role of Nitric Oxide in Endothelium-Dependent Responses

At baseline, assessment of microvascular responses to ACh were repeated in a separate group of WT and MSK1/2 KO mice (*n* = 8 in each group) following pre-treatment with the non-selective inhibitor of nitric oxide synthase (NOS), N (G)-nitro-L-arginine methyl ester hydrochloride (L-NAME, Sigma-Aldrich, St. Louis, United States). These experiments explored the importance of NO in endothelium-dependent dilatation in our model of inflammation. L-NAME was administered by intraperitoneal injections (i.p.) (20 mg/kg in deionized water). 

### 2.9. Assessment of Endothelium-Independent Responses

Following pre-constriction with PE, a 2% solution of sodium nitroprusside (SNP) (Sigma-Aldrich) was iontophoresed on the opposite flank of the animal using a 100 μA cathodal current for 10 min to test endothelium-independent vasodilation. To reduce the amount of time that each animal was under general anaesthetic and to limit the amount of drug exposure, responses to SNP were only measured at the end point of the study. 

### 2.10. Maximum Vasodilator Response to Localised Skin Heating

A skin heating probe (SH02™ Skin Heating Unit and SHP3 probe, Moor Instru-ments, Axminster, UK) with a total surface area of 3.2 cm^2^ was attached to the flank using double sided adhesive rings. Baseline measurements of skin perfusion were taken for 5 min, followed by localized heating of the skin at a rate of 1 °C/min until a maximum temperature of 44 °C was achieved. This was maintained for 10 min and the maximum vasodilator response was measured.

### 2.11. Blood Sampling

Blood was collected by tail bleeds at baseline and by cardiac puncture at study end point into heparinized microtubes (BD Microtainer ^TM^ Lithium Heparin, Franklin Lakes, NJ, USA) to produce plasma by centrifugation. Plasma was stored at −80 °C until analysed.

### 2.12. Plasma Cholesterol

Total cholesterol was measured using a colorimetric assay (Biovision Inc, catalog#: K603–100, Milpitas, United States) high-density lipoprotein (HDL) and low-density/very low-density lipoproteins (LDL/vLDL) fractions were quantified by using a color metric assay (Abcam, Product code: ab655390) as detailed in the manufacturer’s instructions.

### 2.13. Cytokine Analysis

Plasma was analyzed using Bio-Plex^®^ Precision Prokits^TM^ from BIO-RAD laboratories, Hercules, United States, for IL-1α, IL-6, IL-10, TNF-α, E-selectin as detailed in the manufacturer’s instructions. This technique utilizes magnetic beads for multiplex quantification of cytokines in a single sample.

#### Cell Culture

C57BL/6 mouse primary dermal microvascular endothelial cells (Cell Biologics, Chicago, IL, USA) were seeded into tissue culture plates at equal densities in complete mouse endothelial cell medium with the supplement kit (Cell Biologics, Chicago, IL, USA). Cells were stimulated as indicated with 100 ng/mL LPS, 400 ng/mL phorbol 12-myristate-13-acetate (PMA), 10µg/mL anisomycin (aniso), recombinant mouse IL-1α (5 ng/mL) and recombinant mouse TNF-α (10 ng/mL). Cells were pre-treated for 1 hour with inhibitors prior to stimulations as described, with inhibitors purchased in-house from MRC PPU Reagents and Services (https://mrcppureagents.dundee.ac.uk/, last accessed on 5 August 2021) and included the ERK inhibitor PD184352 (10 µM), p38 inhibitor VX-745 (1 µM), MSK1/2 inhibitor SB747581A (10 µM), and MK2 in-hibitor PF36444022 (10 µM). 

### 2.14. Western Blotting

After stimulations, cells were lysed (50 mM Tris-HCl (pH 7.5), 1 mM EGTA, 1 mM EDTA, 1 mM sodium orthovanadate, 50 mM sodium fluoride, 1 mM sodium pyro-phosphate, 10 mM sodium glycerophosphate, 0.27 M sucrose, 1% (*vol*/*vol*) Triton X-100 with Leupeptin 5 mg/mL, Aprotinin 5 mg/mL, PMSF 200 mM, 0.1% β-mercapto-ethanol and bromophenol blue), boiled for 10 min, and passed through a 25nG syringe needle. Thirty µg of lysate were separated on SDS/PAGE with a 10% polyacrylamide gel. Gels were transferred onto nitrocellulose membrane (Amersham, GE-Health Care Life Sciences, Illinois, United States) and blocked in 5% milk-TBS-T (50 mM Tris pH7.5, 150 mM NaCl, 0.1% (*v*/*v*) tween-20) for 1 hour. Subsequently, membranes were probed with primary antibodies overnight (18 h) at 4 °C using a 1 in 1000 dilution in 5% milk-TBS-T. The antibodies used were phospho-CREB at Ser133, phospho-p38 at Thr180/Tyr182, p38-MAPK, phospho-p44/42 MAPK (Erk1/2) at Thr202/Tyr204, p44/42 MAPK (Erk1/2), phospho-MAPKAPK-2 at Thr344, MAPKAPK-2, phospho-HSP27 at Ser 83 (all from Cell Signalling Technologies, Danvers, United States) (Appendix A). In-house anti-sheep MSK1 was used at 5 µg/mL, acquired from the MRC PPU Reagents and Services (https://mrcppureagents.dundee.ac.uk/, last accessed on 5 August 2021). Membranes were probed with appropriate secondary antibodies used at 1 in 10,000 in 5% milk-TBS-T (anti-sheep or anti-rabbit, ThermoFischer) for 1 hour at room temperature, washed and detection was with Clarity ECL reagents from BioRad (Hercules, CA, USA) and imaged on a Licor Odyssey Fc system.

### 2.15. Statistical Analysis

Statistical analysis was performed on SPSS (Chicago, IL, United States) statistical package (v22/24/26). Data are expressed as group means ±SE. 

Quantitative variables were tested for normality in GraphPad Prism (v9) (San Diego, CA, USA) using, Anderson-Darling test, Shapiro-Wilk test, and the D’Agostino and Pearson test. Normally distributed data containing two group comparisons were compared using two tailed unpaired or paired students *T*-tests. Two or more group comparisons were made by one- or two-way ANOVA with post-hoc Bonferroni correction. Associations were assessed by Pearson’s correlation. Multiple linear regression was performed with dependent variables (age, sex, BMI, systolic blood pressure, and diastolic blood pressure). The null hypothesis was rejected at *p* < 0.05.

### 2.16. Study Approval

Patients provided written, informed consent for investigations, which conformed to the principles outlined in the Declaration of Helsinki and local ethical approval was granted by review boards at the University of Dundee.

## 3. Results

### 3.1. MSK1/2 Gene Expression in ST-Elevation Myocardial Infarction (STEMI) Patients 

MSK1 and MSK2 gene expression was significantly lower in PBMCs from STEMI patients when compared to healthy volunteer (HV) controls (STEMI MSK1 (*p* < 0.0001) and MSK2 (*p* < 0.001) vs. HV) (Figure 1A,B, respectively). 

We have previously reported elevated expression of inflammatory cytokines and microvascular dysfunction in patients suffering from rheumatoid arthritis [34] and in an animal model of systemic lupus erythematosus [32], supporting a role for systemic inflammation and innate immune signalling in endothelial-dependent vascular impairment. 

We assessed endothelial function by measuring FMD in our STEMI patients and found they had significantly lower vascular responses compared with HV controls (3.7 ± 1.3% vs. 6.8 ± 1.9%, *p* < 0.0001) (Figure 1C). Additionally, the endothelium-independent response to GTN was significantly lower in STEMI patients compared with HV (STEMI 8.2 ± 0.7% vs. HV 17.8 ± 1.3%, *p* < 0.001). The percentage change in velocity time integral (AUC), as an indicator of microvascular function, before and after occlusion was significantly lower in STEMI patients compared with HV controls (STEMI 415 ± 39% vs. HV 540 ± 39%, *p* < 0.05).

In a linear regression model including age, sex, body mass index (BMI), systolic and diastolic blood pressure, FMD was found to be independent to MSK1 gene expression (β = 0.436, *p* < 0.05) (Figure 1D), MSK 2 gene expression (β = 0.340, *p* < 0.01) (Figure 1E) and BMI (β = −0.280, *p* < 0.05). Additionally, the GTN response was found to be independent to MSK2 gene expression (β = 0.377, *p* < 0.05) and BMI (β = −0.315, *p* < 0.05). The percentage change in velocity time integral (microvascular function) was independent of MSK1 gene expression (β = 0.341, *p* < 0.05) and MSK2 gene expression (β = 0.354, *p* < 0.01), suggesting a role for MSK1/2 in endothelium-dependent microvascular function.

### 3.2. MSK1 Kinase in Endothelial Cells

To further validate these observations, we studied MSK1 activation in dermal skin microvascular endothelial cells. We stimulated endothelial cells in vitro with pro-inflammatory factors: lipopolysaccharide (LPS), phorbol 12-myristate 13-acetate (PMA), recombinant mouse IL-1α, TNF-α, or p38 MAPK activator anisomycin. MSK1 was detected in skin microvascular dermal endothelial cells (Appendix A). Anisomycin, PMA, and TNF-α activated p38 MAPK and phosphorylated CREB (Appendix A) and we hypothesized that MSK1/2 kinases may be involved in the development of endothelial dysfunction.

### 3.3. MSK1/2 KO Mice Show Altered Plasma Cytokines but Similar Vascular Responses to WT Mice at Study Baseline

To determine the role of MSK1/2 in vascular function, we utilized MSK1/2 KO mice and in vivo LDI. This translational approach allowed vascular function testing longitudinally in the same animal population. Baseline body weights at 16 weeks of age were significantly greater in WT than in MSK1/2 KO mice (28 ± 1 g vs. 26 ± 1 g, respectively, *p* < 0.01) (Appendix A). Additionally, they displayed elevations in plasma IL-1α (*p* < 0.01), IL-6 (*p* < 0.001), soluble E-selectin (*p* < 0.01), and lower levels of IL-10 (*p* < 0.0001), when compared against WT mice (Figure 2A–D). Plasma levels of TNF-α and total plasma cholesterol levels were similar between MSK1/2 KO and WT control animals (Figure 2E–F). MSK1/2 KO mice had similar vascular responses to WT littermates (Figure 3A), suggesting that the degree of systemic inflammation and potential duration did not alter vascular function. Endothelium-dependent vascular responses are mediated by the bioavailability of nitric oxide (NO); therefore, we studied whether MSK1/2 KO mice exhibited changes in NO bioavailability.

### 3.4. L-NAME Inhibits Endothelium Dependent Vasodilation in MSK1/2 KO Mice at Baseline

At baseline, administration of the endothelial NO synthase (NOS) inhibitor, L-NAME produced a significant reduction in the peak endothelial dependent vascular response in MSK1/2 KO and WT control mice (320 ± 56 to 257 ± 60 AU, *p* < 0.01 in WT mice and 318 ± 60 to 243 ± 65 AU, *p* < 0.01 in MSK1/2 KO mice). This demonstrated that endothelium-dependent vascular responses were mediated by NO in MSK1/2 KO and WT mice at baseline.

### 3.5. MSK1/2 KO Animals Show Endothelial Dysfunction and Loss of Nitric Oxide

WT mice on a standard chow diet displayed similar microvascular responses over the study period, (Figure 3A) demonstrating preserved NO-mediated vasodilatation. WT animals fed a cholesterol diet show attenuated vascular responses, in agreement with our previous studies [33]. MSK1/2 KO mice on a standard chow or cholesterol enriched diet showed significant attenuation of endothelium-dependent responses over time (*p* < 0.001) (Figure 3A). MSK1/2 KO on either diet showed no significant change to L-NAME at study week 12 (chow-fed 226 ± 18 AU versus 251 ± 18 AU, *p* > 0.05 and cholesterol-fed 235 ± 19 AU versus 237 ± 38, *p* > 0.05), indicating that diminished NO bioavailability was responsible for the attenuated vascular responses in MSK1/2 KO mice.

### 3.6. Maximum and Endothelium-Independent Responses 

To better understand maximal vasodilator capacity in the skin microcirculation measurements using our in vivo setup, we utilized localized skin heating to 44 °C. We found similar responses amongst the groups, suggesting that cholesterol feeding in WT and MSK1/2 deficiency in mice (cholesterol and chow groups) affects endothelial function specifically without affecting generalized smooth muscle function (Figure 3B). Importantly, localized skin heating of the skin microvasculature provides a drug free measurement of vascular function in rodents in combination with LDI. We confirmed endothelium-independent microvascular responses at study end-point by iontophoresis of SNP and found that responses were not significantly different amongst the study groups. This suggests localized damage to the endothelium in MSK1/2 KO animals and WT-cholesterol fed mice (Figure 3C).

### 3.7. Plasma Cytokines Are Altered in MSK1/2 KO Mice and Correlate with Vascular Function

At the end of the experiment, MSK 1/2 KO cholesterol-fed mice had significantly greater plasma IL-1α (Figure 4A) when compared to WT-cholesterol fed mice. Soluble E-selectin (Figure 4B), TNF-α and IL-6 were significantly greater in MSK1/2 KO chow and cholesterol-fed and MSK1/2 KO cholesterol-fed mice at study end-point, when compared to WT-controls (Figure 4C,D). Cholesterol feeding in MSK 1/2 KO mice diminished IL-10 when compared to WT-cholesterol fed animals at study end point (Figure 1E).

Endothelium-dependent microvascular responses and localized skin heating at study baseline showed no significant correlations with the blood derived parameters including cytokines and cholesterol. In contrast, a number of significant correlations were observed between blood markers and endothelium-dependent microvascular responses at study end point. Importantly, these correlations were not present with SNP iontophoresis or with the response to localized skin heating. Peak endothelium-dependent responses at study endpoint (24 weeks) significantly correlated with IL-1α (R^2^ = −0.514, *p* < 0.001), IL-6 (R^2^ = −0.581, *p* < 0.001), IL-10 (R^2^ = 0.651, *p* < 0.001), TNF-α (R^2^ = −0.617, *p* < 0.001), and E-selectin (R^2^ = −0.424, *p* < 0.001). From the univariate correlations above, these were entered into a stepwise linear regression model. Independent determinants of endothelium-dependent responses were IL-10 (β= 0.483, *p* < 0.001) and IL-1α (β= −0.286, *p* < 0.001).

### 3.8. MSK1/2 KO-Cholesterol Fed Mice Are More Susceptible to Dyslipidaemia

MSK1/2 KO-cholesterol fed mice had significant greater dyslipidaemia, when compared to WT-cholesterol fed mice (Figure 4F). 

### 3.9. Body Weights

WT-chow (baseline 27 ± 1 g vs. week 24 32 ± 1 g, *p* < 0.001), WT-cholesterol (baseline 30 ± 1 g vs. week 24 36 ± 1 g, *p* < 0.001), KO-chow (baseline 27 ± 1 g vs. week 24 32 ± 1 g, *p* < 0.001) and KO-cholesterol (baseline 26 ± 1 g vs. week 24 35 ± 1 g, *p* < 0.001) fed mice significantly increased in body weight over the study duration. End-point body weights were significantly greater in cholesterol fed WT and MSK 1/ 2 KO mice (*p* < 0.001) when compared to control chow fed mice (Appendix A).

### 3.10. Preservation of Endothelium-Dependent Responses and Nitric Oxide Activity in MYD88 KO-Cholesterol Fed Mice

We have previously demonstrated that MSK1/2 are activated after TLR stimulation [19]. TLR activation involves the recruitment of MyD88 to initiate cytoplasmic signalling. MyD88 KO mice has been previously shown to reduce atherosclerotic plaque formation [35,36]. However, the role of MyD88 in endothelial function has not previously been reported in vivo. To ascertain whether TLR activation induces endothelial dysfunction in response to dietary cholesterol, we fed MyD88 KO mice a cholesterol rich diet. MyD88 KO (27 ± 1 g) mice were significantly heavier than WT (25 ± 1 g) animals at study baseline (*p* < 0.01) (Appendix A). MyD88 KO-cholesterol fed mice were resistant to endothelial dysfunction in vivo (Figure 5A,B). MyD88 KO-cholesterol fed mice had significantly lower plasma levels of IL-6 and IL-1α when compared with WT-cholesterol control mice (Figure 5C,D). In agreement with previous reports, [35] we show that MyD88-cholesterol fed animals display exaggerated dyslipidaemia versus WT-cholesterol fed mice (Figure 5F,G). This supports the view that cholesterol feeding in mice induces microvascular dysfunction and systemic cytokine expression through a TLRs-MyD88-MSK1/2 signaling pathway. Animals in all groups significantly increased in body weight over the study duration: WT-chow (baseline 26 ± 1 g vs. week 20 29 ± 1 g, *p* < 0.001), WT-cholesterol (baseline 25 ± 1 g vs. week 20 30 ± 1 g, *p* < 0.001) KO-chow (baseline 27 ± 1 g vs. week 20 32 ± 1 g, *p* < 0.05) and KO-cholesterol (baseline 26 ± 1 g vs. week 20 30 ± 1 g, *p* < 0.001) but there were no differences amongst the groups for body weights at study end point (*p* > 0.05) (Appendix A).

### 3.11. The Role of MAPKAP 2/3 in Vascular Dysfunction and Plasma Cytokine Expression

As p38α activates MAPKAPK2 and 3 in addition to MSK1/2. We investigated the role of MAPKAPK2 and 3 using MAPKAPK2/3 KO mice. These two kinases are important messengers in the inflammation cascade, and their involvement is mainly in the regulation of gene expression [21,37]. MAPKAP 2/3 have shown potential as a therapeutic target in a KO mouse models, through abrogated cytokine release and less atherosclerotic plaque formation in a collagen induced rheumatoid arthritis animal model [22,23].

We assessed whether MAPKAP 2/3 KO mice were susceptible to endothelial dysfunction using our in vivo model. There were no significant differences for vascular function at study baseline between WT and MAPKAP 2/3 KO mice for endothelium-dependent vasodilatation (Figure 6A) or maximal vasodilator dilator capacity to localized skin heating. WT (27 ± 1 g) animals were significantly heavier than KO mice at study baseline (23 ± 1 g) (12 weeks of age) (*p* < 0.001).

### 3.12. Preservation of Nitric Oxide in MAPKAP 2/3 KO Mice in Response to Dietary Cholesterol

MAPKAP 2/3 KO-cholesterol fed mice had similar endothelium-dependent responses to WT-chow fed mice at study weeks 8 and 12, suggesting that MAPAKP 2/3 signalling is involved in dietary cholesterol mediated vascular dysfunction. MAPAKP 2/3 KO-cholesterol fed animals had greater endothelium-dependent vasodilation at study week 20 measurements when compared with WT-cholesterol fed mice. However, these observations did not meet statistical significance for multiple comparisons (*p* > 0.05) (Figure 6A).

L-NAME administration to MAPKAP 2/3 KO-cholesterol fed mice at 20 weeks induced significant attenuation in the peak endothelium-dependent response compared with the response without L-NAME (*p* < 0.001), demonstrating preservation of NO bio-availability. These data show that TLR-MyD88-MSK1/2 activation diminishes NO bioavailability and mediates endothelial dysfunction.

## 4. Discussion

In this study, we show decreased gene expression of MSK1 and MSK 2 in PBMCs from patients presenting with STEMI. This finding correlates with attenuated endothelium-dependent macrovascular (FMD) and microvascular (velocity time integral) function. In mice, MSK1/2 deficiency caused elevated systemic pro-inflammatory cytokine expression and decreased anti-inflammatory cytokine expression with significant impairment in endothelium-dependent vasodilatation. Blood cytokine levels correlated negatively with microvascular function in mice. Endothelium-dependent skin microvascular responses are dependent on the bioavailability of NO. Our studies using L-NAME showed that endothelium-dependent responses were, in part, mediated via the actions of NO, and therefore, a reduction in endothelium-dependent responses in MSK1/2 deficient mice was possibly mediated via loss of NO bioavailability. In contrast, WT animals fed a normal chow diet over the 24-week study period maintained normal plasma lipids, showed no significant changes in basal levels of inflammatory markers and had preservation of endothelium-dependent vascular responses. Generalized vascular function, tested by iontophoresis of the endothelium-independent vasodilator SNP and maximum vasodilator capacity to localized skin heating was not significantly different amongst the groups, suggesting that early elevation in pro-inflammatory cytokines was affecting vascular function specifically at the level of the endothelium. These data suggest a possible functional role for MSK1/2 in the development and progression of CVD through an interaction between regulation of inflammatory cytokines and endothelial dysfunction. These data support clinical studies, which have linked blood cytokine levels to the development and progression of CVD [5,6,7].

Endothelial cells are not traditionally characterized as components of the innate immune system but they express TLRs, similar to monocytes [9], macrophages [35,38] and neutrophils [39], which are implicated in the development, progression, and regression of vascular disease. Endothelial cells interface with circulating blood and thus alterations in systemic cytokine expression and/or changes in plasma lipids can influence endothelial cell biology. TLRs activation is important in the initiation of pro-inflammatory responses in immune cells and TLR deficiency attenuates the progression of atherosclerotic plaques in hyperlipidemic mice [35]. TLR activation can be initiated by stimuli, which are risk factors for CVD, including oxidized low-density lipoproteins. TLR signalling is complex and involves the recruitment of MyD88, which enables transcytoplasmic signalling and activation of inflammatory cascades via MAPKs, p38, and NF-kB. We demonstrate here that MyD88 is necessary for endothelial cell dysfunction in the face of hypercholesterolemia in vivo. Similarly, TLR-MyD88 effects have been reported in a model of atherosclerosis [35], obesity [36] and diabetes [40]. We believe this is the first description of TLR-MyD88-MSK1/ 2 kinases in the initiation of early vascular disease in an in vivo rodent model relevant to human microvascular impairment in CVD. 

Baseline endothelial-dependent responses were dependent on NO in both WT and MSK1/2 KO mice as shown by the significant reduction following pre-treatment with L-NAME. At study baseline, although we found elevations in pro-inflammatory markers in MSK1/2 KO mice compared with WT mice (increased IL-1α, IL-6, E-selectin and reduced IL-10), we did not see any significant differences in endothelial-dependent responses, perhaps indicating that the relative difference and potential duration of change in cytokine levels was not sufficient enough to impact significantly on vascular function at that time point. The difference in inflammatory cytokines between WT and KO mice at baseline may be mediated by environmental factors e.g., exposure to infectious agents. 

Mice were initially bred and maintained in a barrier facility and subsequently re-housed for vascular testing in a conventional unit, at which they could have been exposed to additional micro-organisms. The onset of CVD and induction of pro-inflammatory gene expression in immune compromised animals has previously been reported in non-pathogen free conditions when animals are challenged with high cholesterol feeding [41]. Previous findings show that high cholesterol feeding induces greater adverse changes in pro-inflammatory cytokines [33], with resultant endothelial dysfunction in WT mice [42]. Interestingly, we found that MSK1/2 KO mice fed a normal chow diet over 24 weeks also displayed elevations in pro-inflammatory cytokines and diminished endothelial-dependent responses. This shows that in the absence of a major CVD risk factor, i.e., elevated plasma cholesterol, mice lacking MSK1/2 still develop significant endothelial dysfunction, to a degree similar to that seen with cholesterol feeding. The exact mechanism for this dysfunction requires further scrutiny. Interestingly, despite a greater reduction in NO bioavailability in MSK1/2 KO mice, endothelium-dependent responses in the presence of L-NAME were not as low as responses in WT cholesterol-fed mice post L-NAME. It is possible that the higher level of perfusion in the MSK1/2 KO mice might be related to a compensatory increase in ACh-induced production of endothelium-derived hyperpolarizing factor [43].

We found significant correlations between microvascular responses and levels of cytokines at 24 weeks, which demonstrates that our methodology for assessment of microvascular function in the skin is sensitive and reflective of systemic inflammation and is similar to what we have reported in patients [34]. IL-10 proved to be the strongest independent determinant of endothelium-dependent response in a multivariate regression model with IL-1α, also proving to be an independent determinant. IL-1α is unregulated in cholesterol fed mice and can stimulate atherosclerosis [33,44]. IL-1α is synthesized by activated endothelial cells and is released early in the atherosclerotic process and mediates the expression of cell adhesion molecules responsible for the capture, rolling, and subsequent transmigration of immune cells into the sub-endothelial space [45]. IL-1α levels are upregulated in cholesterol fed animals. These regulatory roles of IL-1 are associated with a number of physiological events including macrophage activation, endothelial proliferation [46], myocardial cell damage, and endothelial dysfunction through increased oxidative stress and inflammation [47,48]. IL-6 has been linked to an increased risk of adverse cardiovascular events [49,50] and can be synthesized by a number of cells including macrophages. IL-6 inhibits endothelial NOS (eNOS) activation and attenuates vasodilation by increasing the half-life of caveolin-1, resulting in more eNOS binding, leading to reduced bioavailability of NO [51]. Thus, the pre-disposition of MSK1/ 2 KO mice to produce increased levels of IL-6 may directly mediate the observed endothelial dysfunction through altered eNOS signaling. IL-10 has a potential protective role in atherosclerosis and can be produced by macrophages and inhibits pro-inflammatory cytokine expression through a JAK/STAT3 dependent pathway [52]. Ananieva et. al. [19] reported that the increased production of IL-6, and to a lesser extent TNF, was dependent on the ability of MSKs to regulate IL-10 production. IL-10 KO mouse aortas display significantly blunted endothelium-dependent responses by a reduction in eNOS expression [53]. Exogenous IL-10 treatment prevents intimal hyperplasia in response to carotid injury in otherwise healthy mice [54]. IL-10 is anti-atherogenic and enhances uptake and efflux of cholesterol, lowering levels of cell death and the progression of atherosclerotic lesions [55]. 

Taken together, the increased levels of IL-1α, IL-6, and TNF-α in MSK1/2 KO mice could explain the observed endothelial-dysfunction in this study, although the mechanisms of diminished NO bioavailability may be multi-modal and requires further scrutiny. It has previously been reported that MSK1/2 KO mice have decreased production of IL-1RA [17]. IL-1RA KO mice display a significant elevation in plasma cholesterol due to impaired cholesterol efflux through altered conversion of hepatic cholesterol to bile acids. This is mediated by the reduced expression of 7α-hydroxylase (CY7A1), the rate limiting step in bile acid synthesis [56]. MSK1/ 2 KO mice require further study to assess the exact mechanism of altered cholesterol homeostasis.

## 5. Limitations

Our STEMI patients were recruited to the study varying in 12–100 h post PCI and there was a bias towards male patients. Here, we describe the expressions of MSK 1/2 PBMCs in STEMI patient and show lower expression and a causal relationship between attenuated endothelial function in patients following AMI. However, PBMCs are highly heterogenous and MSK 1/2 expression may vary considerably between peripheral blood immune cell populations, such as neutrophils, monocytes, and lymphocytes, which show distinct activation patterns following AMI. Our PBMCs samples from STEMI patients showed invariant expression for actin-beta under our experimental conditions. However, actin-beta expression may vary amongst individual PBMCs populations and therefore a larger panel of housekeeper genes may be necessary in future studies looking to explore the relationship between MSK 1/2 expression in distinct immune cell populations. Our study focused on the role of MSK 1/2, which are activated downstream of the ERK1/2 and p38 pathways; a parallel pathway is NF-kB, which has shown roles in endothelial cell activation and immune cell activation. It was not be possible to address all the possible pathways activated by MyD88 using the genetic approaches we have used in this study but future studies looking at NF-kB may benefit from the translational microvascular techniques we have employed here. Global transgenic KO mouse models were used in these studies and do not allow us to ascertain which individual cell types may be responsible for the observed endothelial-dependent vascular dysfunction in the MSK 1/2 KO mouse model. Targeted deletion using tissue specific models for endothelial and immune cell loss of MSK 1/2 will provide detailed information on how endothelial-dependent vascular dysfunction is mediated in the absence of MSK 1/2. 

## 6. Conclusions

In conclusion, we have shown that lack of MSK1/2 might be contributing to the inflammatory response following STEMI, and we have used an in vivo model to examine the longitudinal changes in the skin microcirculation of WT and KO mice prone to inflammation. MSK1/2 plays an important role in limiting pro-inflammatory signalling downstream of TLRs, and here we show that deletion of MSK1/2 produces a marked inflammatory state with consequent early endothelial dysfunction and reduced NO bioavailability. Targeting the MSK1/2 pathway with agents that activate MSK1 and MSK 2 might provide a useful therapeutic option for combating inflammation-induced endothelial dysfunction, measured by attenuated vasodilatation in the current studies.

## Figures and Tables

**Figure 1 ijms-22-08655-f001:**
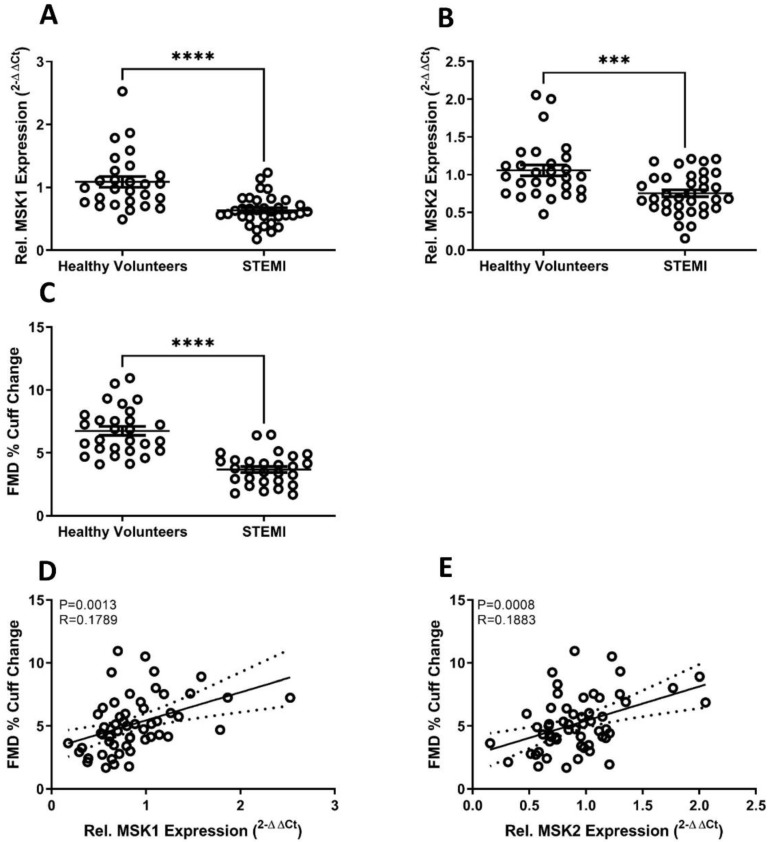
MSK1/2 expression and endothelial function are attenuated following acute myocardial infarction. Relative MSK1 (**A**) and MSK2 (**B**) gene expression in peripheral blood mononuclear cells from healthy volunteers and patients presenting with ST-segment elevation acute myocardial infarction (STEMI). (**C**) Flow mediated dilation (FMD) (% cuff change) in STEMI patients and healthy volunteers. Pearson’s Correlation between FMD response and MSK1 (**D**)/ MSK2 (**E**) gene expression. Healthy volunteers *n* = 28 STEMI *n* = 34. Values are group mean ± SE. (**A**–**C**) Unpaired *T*-tests. *** *p* < 0.001, **** *p* < 0.0001. E/F Person’s correlation, dotted line represents 95% confidence interval.

**Figure 2 ijms-22-08655-f002:**
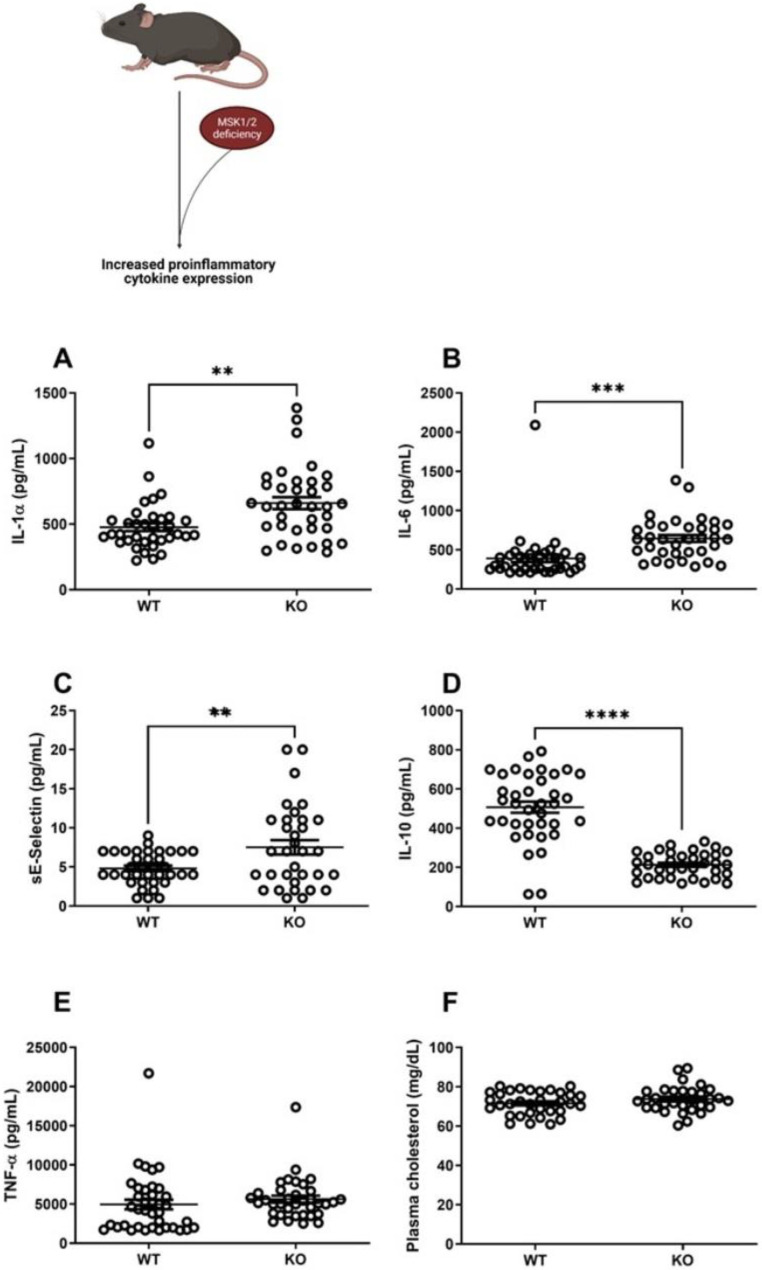
Plasma cytokine and cholesterol measurements in MSK1/2 knock-out (KO) and wild-type (WT) mice. (*n* = 36) wild-type and MSK1/2 knock-out (*n* = 38). Plasma levels of inflammatory markers in mice (pg/mL) at study baseline: (**A**) IL-1α, (**B**) IL-6, (**C**) sE-Selectin, (**D**) IL-10, (**E**) TNF-α, (**F**) total plasma cholesterol (mg/dL). Values are group mean ± SE. (**A**–**F**) Unpaired *T*-tests. ** *p* < 0.01, *** *p* < 0.001, **** *p* < 0.001.

**Figure 3 ijms-22-08655-f003:**
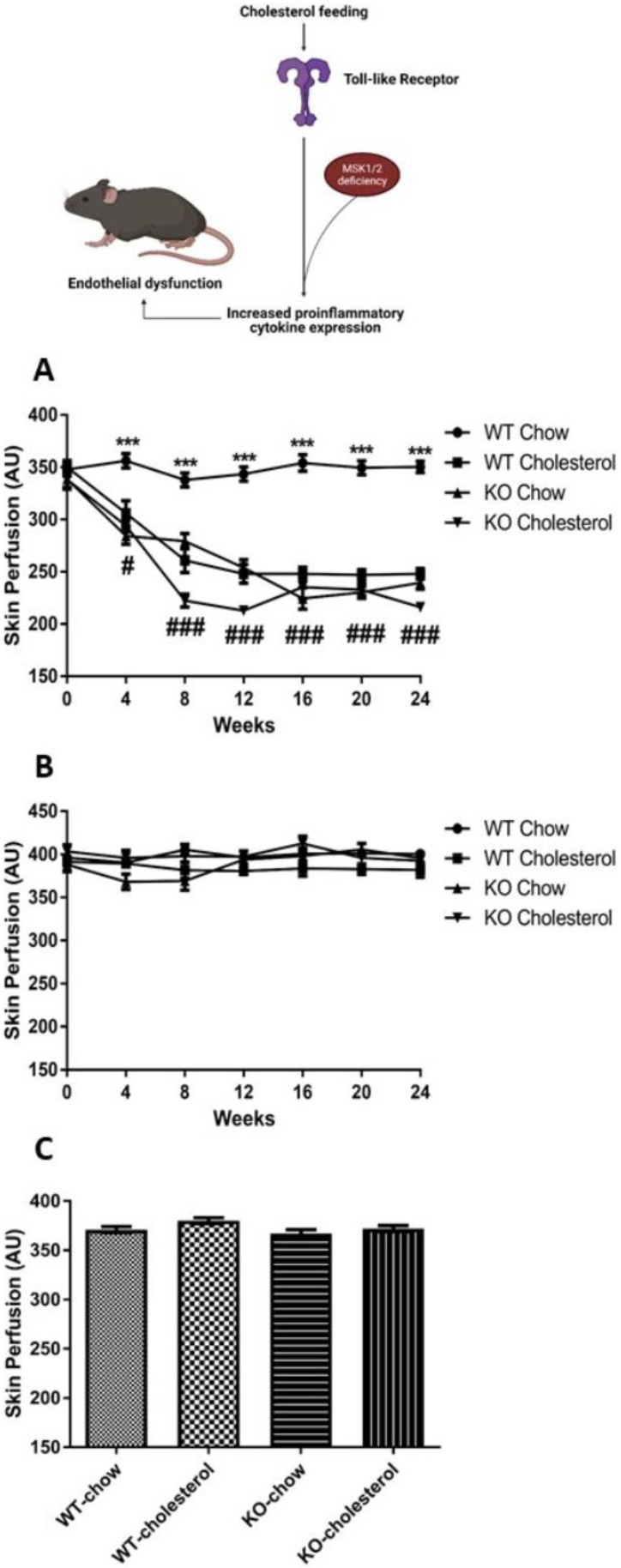
Microvascular responses in MSK1/2 knockout and wild-type mice. (**A**) Endothelial-dependent microvascular responses in arbitrary units (AU) over 24 weeks and (**B**) localized 44 °C heating (over 24 weeks) and terminal endothelial-independent (**C**) in wild-type (WT)-chow (*n* = 19), WT-cholesterol (*n* = 19), MSK1/2 knock-out (KO)-chow (*n* = 18) and MSK1/2 KO-cholesterol (*n* = 18) fed animals. Longitudinal changes were analyzed by ANOVA for repeated measures followed by post-hoc Bonferroni correction. Values are group means ± SE. *** *p* < 0.001 comparing genotype responses between WT-chow and # *p* < 0.05, and ### *p* < 0.001, comparing diet with WT-chow.

**Figure 4 ijms-22-08655-f004:**
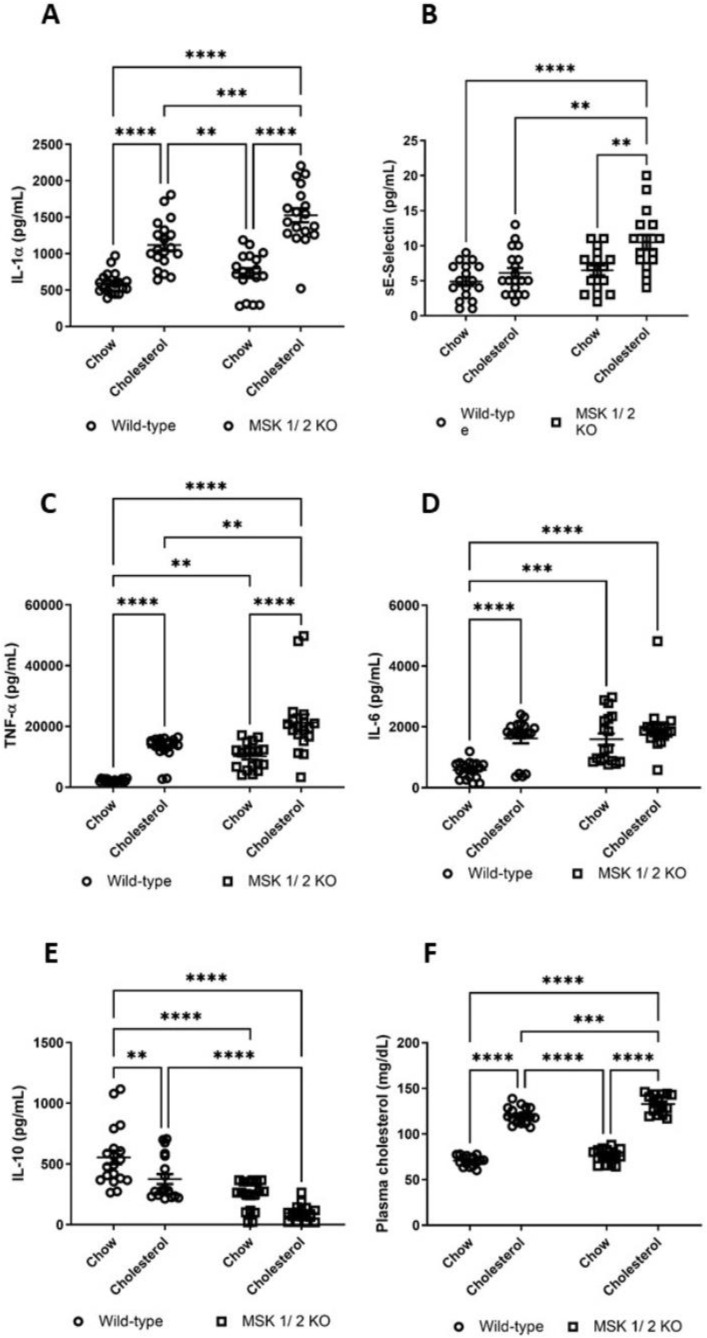
Plasma cytokine and cholesterol measurements in MSK1/2 knockout and wild-type mice. Wild-type (WT)-chow (*n* = 19), WT-cholesterol (*n* = 19), MSK1/2 knock-out (KO)-chow (*n* = 18) and MSK1/2 KO-cholesterol (*n* = 18) fed mice (pg/mL). (**A**) IL-1α, (**B**) sE-Selectin, (**C**) TNF-α, (**D**) IL-6, (**E**) IL-10, (**F**) total plasma cholesterol (mg/dL). Differences between genotype groups were tested using two way-ANOVA followed with post-hoc Bonferroni correction. Values are group means ± SE. ** *p* < 0.01, *** *p* < 0.001, **** *p* < 0.0001.

**Figure 5 ijms-22-08655-f005:**
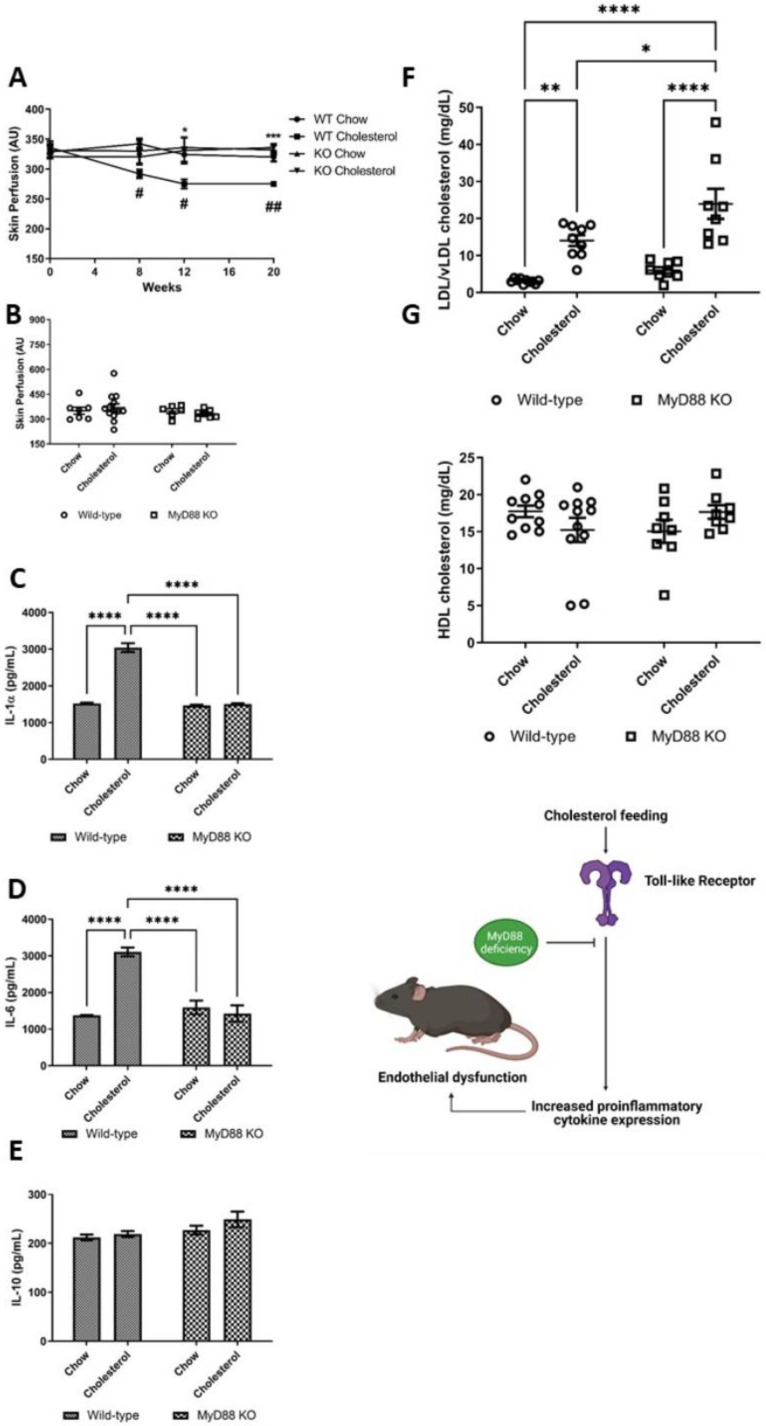
Microvascular responses in MyD88 knockout and wild-type mice. (**A**) Endothelium-dependent microvascular responses in arbitrary units (AU) over 20 weeks and (**B**) terminal endothelium-independent in wild-type (WT)-chow (*n* = 9), WT-cholesterol (*n* = 15), MyD88 knock-out (KO)-chow (*n* = 9) and MyD88 KO-cholesterol (*n* = 8) fed animals. Longitudinal changes were analysed by ANOVA for repeated measures followed by post-hoc Bonferroni correction. (**A**,**B**) * *p* < 0.05, *** *p* < 0.001 comparing genotypes, # *p* < 0.05, ## *p* < 0.01 comparing diet. Plasma cytokines (pg/mL) (**C**) IL-1α, (**D**) IL-6 and (**E**) IL-10 WT-chow (*n* = 10), WT-cholesterol (*n* = 10), MyD88 KO-chow (*n* = 10) and MyD88 KO-cholesterol fed (*n* = 10) mice. Plasma measurements of (**F**) low density lipoproteins and very low density lipoproteins (LDL/vLDL) and (**G**) high density lipoproteins (HDL) in the plasma of WT-chow (*n* = 10), WT-cholesterol (*n* = 11), MyD88 KO-chow (*n* = 8) and MyD88 KO-cholesterol fed mice (*n* = 8) (mg/dL). (**C**–**E**) two-way ANOVA. Values are group means ± SE. * *p* < 0.05, ** *p* < 0.01, **** *p* < 0.0001.

**Figure 6 ijms-22-08655-f006:**
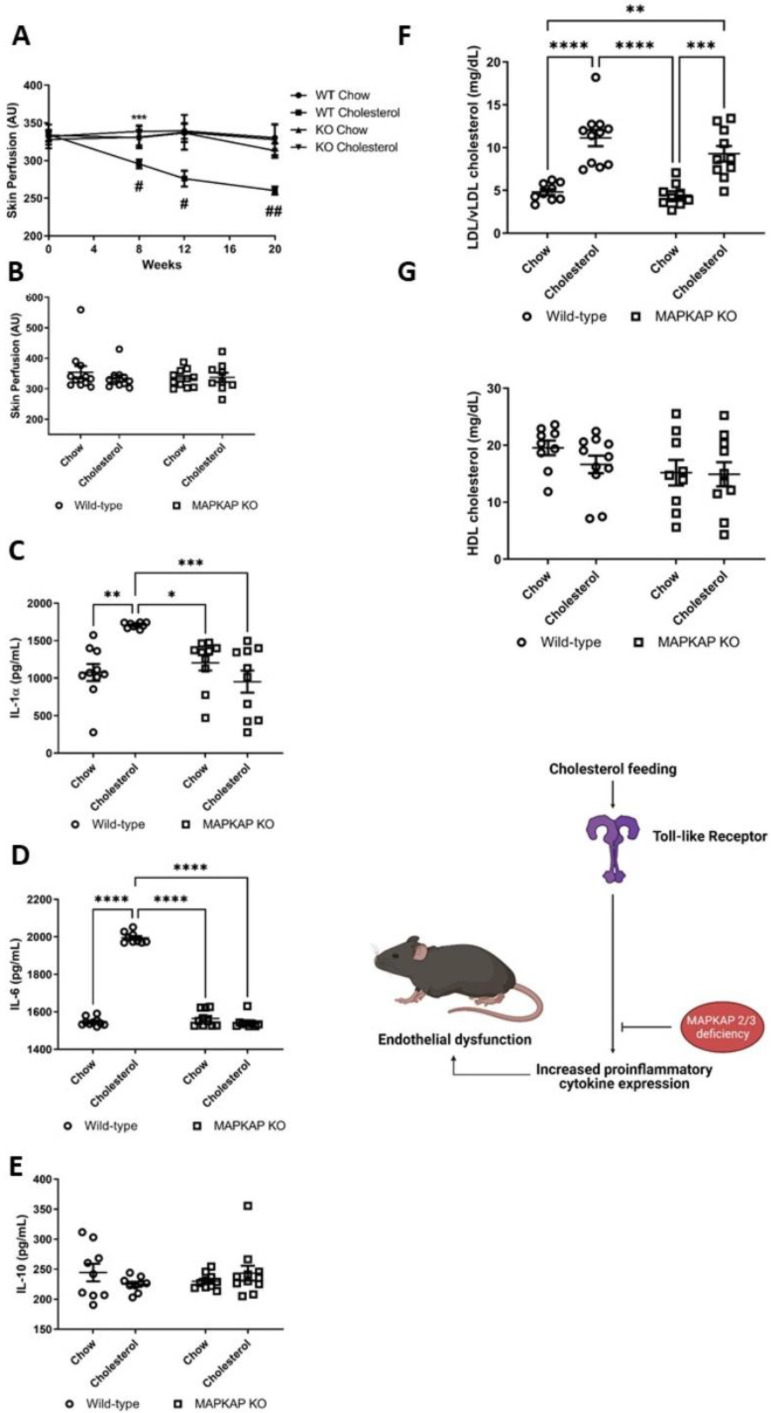
Microvascular responses in MAPKAP2/3 knockout and wild-type mice. (**A**) Endothelium-dependent micro-vascular responses in arbitrary units (AU) over 20 weeks and (**B**) terminal endothelium-independent in wild-type (WT)-chow (*n* = 17), WT-cholesterol (*n* = 12), MAPKAP knock-out (KO)-chow (*n* = 11) and MAPKAP KO-cholesterol (*n* = 13) fed animals. Longitudinal changes were analysed by ANOVA for repeated measures followed by post-hoc Bonferroni correction. (**A**,**B**) *** *p* < 0.001 comparing genotypes, # *p* < 0.05, ## *p* < 0.01 comparing diet. Plasma cytokines (pg/mL) (**C**) IL-1α, (**D**) IL-6, and (**E**) IL-10 in WT-chow (*n* = 10), WT-cholesterol. (*n* = 10), MAPKAP KO-chow (*n* = 10) and MAPKAPKO-cholesterol fed (*n* = 10) mice. Plasma measurements of (**F**) low density lipoproteins and very low-density lipoproteins (LDL/vLDL) and (**G**) high density lipoproteins (HDL) in the plasma of WT-chow (*n* = 10), WT-cholesterol (*n* = 9), MAPKAP KO-chow (*n* = 9) and MAPKAP KO-cholesterol fed mice (*n* = 10) (mg/dL). (**C**–**G**) two-way ANOVA values are group means ± SE. * *p* < 0.05, ** *p* < 0.01, *** *p* < 0.001, **** *p* < 0.0001.

## Data Availability

All supporting data are available by contacting the corresponding authors.

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
