# Peer review of "Mitogen and Stress-Activated Kinases 1 and 2 Mediate Endothelial Dysfunction"

_ijms, 2021, doi:10.3390/ijms22168655_

Round 1

Reviewer 1 Report

In this study, the author would like to emphasize the role of Mitogen and Stress-Activated Kinases 1 and 2 in mediating endothelial dysfunction, the study designs and results are interesting, only some points are suggested making the results more convincing.    

1. In figure 1, the results demonstrated that MSK1/2 gene expression and endothelial function are attenuated following acute myocardial infarction. I suggest the protein level of MSK1/2 expression should be presented in the animal study by IHC or western blot, if there were enough specimens. 

2. In the conclusion of “selective activation of MSK1/2 could reduce pro-inflammatory responses and preserve endothelial function” the expression of pro-inflammatory factors is from what types of cells? Meanwhile, what is the endothelial function is suggested to clear define. 

3. In the description of “MSK1/2 are activated by toll-like receptors and require the adapter protein 40 MyD88 and downstream MAPKAP2/3, which are necessary for cytokine synthesis” , the pathway is working in which types of cell?  

Reviewer 2 Report

 The current manuscript is presenting very interesting data obtained both from ST-elevation myocardial Infarction (STEMI) patients and from a KO animal model to argue the role of  mitogen-activated-protein-kinases (MAPKs) in endothelial dysfunction; this combination of experimental models is the main strong point of the manuscript . Results prove that   mitogen and stress kinase 1/2 (MSK1/2) are decreased in PBMCs in association with endothelial dysfunction. The animal model sustained the human data;  genetic deficiency of MSK1/2  was associated with reduced NO production and increased inflammatory factors expression. The MSK1/2  could constitute a potential therapeutic target for improving endothelial dysfuntion

Manuscript is very interesting and should be published as it is!

Reviewer 3 Report

Dear authors,

This study tries to analyze the role of mitogen and stress-activated kinases 1 and 2 in endothelial dysfunction, and more specifically, in vascular disease. The manuscript is well structured and written, and the introduction provide sufficient background and include relevant references. However, many modifications and information are necessary to add before publication.

In material and methods section, a lot of information needs to be added. For example, the authors should explain with more details the RT-qPCR method, adding housekeeping gene used and why used that and not other. In this section, a table with primers sequences would be good. In “mouse models” section, the authors should indicate the genetic background of animals used. In “cytokine analysis”, the authors should explain the method used (elisa?) and the commercial kits used for analysis. In “western blot” section, the first paragraph is not of Western Blotting procedure. Perhaps, the authors should add a cell culture section and explain it in more detail. Furthermore, the western blot procedure must explain with more detail (time to blocked, time to incubation of secondary antibodies, what secondary antibodies used, etc). A table with antibodies used could be useful. In “statistical analysis” section, the authors indicate that they have used two statistical packages, why?, and before to use a parametric tests, as t-student and ANOVA, the authors must verify that the data are distributed in a normal distribution and that present homoscedasticity.

In the results, the first paragraph is material and methods, not results. The table 1 is absent. The authors indicate p-values<0.001 as significant, but in the material and methods section, the p-value for statistical analysis was <0.05. A lower p-value do not give more information. The authors should consolidate the data in this regard.

These are some of the fixes that should be made at work, but there are more.

Round 2

Reviewer 3 Report

Dear authors,

Although the manuscript has improved considerably, it still has shortcomings that need to be addressed.

  • Section 2.4: The authors still do not explain in detail the procedure for RT-qPCR. They must specify the PCR program used. On the other hand, they should indicate why the 18S and GADPH housekeeping were discarded (data obtained from these housekeeping genes). On the other hand, they must include information about the results obtained in the extracted RNA (purity and quality, how it has been measured, what values have been taken into account as valid). They should also include how much cDNA has been used in each PCR reaction.
  • Section 2.15: Why have you used two statistical packages? The SPSS program is capable of performing any statistical analysis.
  • line 257: A lower p-value does not indicate a lower probability. A hypothesis test has two possible hypotheses, the null hypothesis or the alternative. The p-value is the value that is taken into account in the statistical analysis to reject or not the null hypothesis, so if a p-value <0.05 has been taken as a reference value, all values <0.05 will have the same meaning. Classic statistics do not reflect probabilities.
    Authors should review the meaning of the p-value and the hypothesis tests.

Round 3

Reviewer 3 Report

Dear authors,

The manuscript has improved considerably. However, you still don't understand the meaning of a p-value. When performing a statistical analysis, the p-value <0.05 indicates statistical significance, as you have written. However, lower values (for example, p-value <0.001) do not indicate "higher significance", since this concept does not exist in classical statistics. Please, unify the results indicated in the study, indicating only if the p-value has been less than 0.05 or not.
